# On-Demand Energy Transfer and Energy-Aware Polling-Based MAC for Wireless Powered Sensor Networks

**DOI:** 10.3390/s22072476

**Published:** 2022-03-23

**Authors:** Mingfu Li, Ching-Chieh Fang, Huei-Wen Ferng

**Affiliations:** 1Department of Electrical Engineering, School of Electrical and Computer Engineering, College of Engineering, Chang Gung University, Guishan District, Taoyuan 33302, Taiwan; m0921003@cgu.edu.tw; 2Department of Computer Science and Information Engineering, National Taiwan University of Science and Technology, Taipei 10607, Taiwan; hwferng@mail.ntust.edu.tw

**Keywords:** wireless powered sensor network (WPSN), on-demand wireless energy transfer, energy-aware medium access control (MAC) protocol, polling protocol, Internet of things (IoT)

## Abstract

To improve the performance of the wireless powered sensor network (WPSN), this paper proposes a frequency division duplex (FDD)-based on-demand energy transfer protocol and an energy-aware polling-based medium access control (MAC) protocol, called composite energy and data first (CEDF), by using the numbers of data packets and energy packets to determine polling priorities. The performance of the proposed MAC protocol, i.e., CEDF, along with the on-demand energy transfer protocol was evaluated through simulations, with comparison to the closely related protocols such as the round robin (RR) and data first (DF) polling protocols. Compared with RR and DF, our proposed CEDF performs much better in terms of throughput, data packet loss rate, and delay. Additionally, the doubly near–far problem in WPSNs under our proposed on-demand energy transfer protocol and CEDF was investigated to come up with good solutions to alleviate such a problem.

## 1. Introduction

In the Internet of things (IoT), there are numerous low-power wireless sensors for detecting and acquiring environmental data [1,2,3]. Such wireless sensors have also been popularly employed for healthcare applications [4,5]. Actually, conventional wireless sensors with replaceable batteries may encounter the energy depletion problem, incurring an additional maintenance cost in replacing batteries. Applying the radio-frequency (RF)-based wireless energy transfer technology may solve such a dilemma [6,7,8,9,10,11]. However, the wireless charging rates for RF-based wireless powered sensors are usually unstable and low because of the time-varying characteristics of wireless channels and health concerns. To improve system performance of a wireless powered sensor network (WPSN), multi-antenna beamforming technologies [12,13,14,15], multi-source energy transfer schemes [10,16,17,18], and energy-aware medium access control (MAC) protocols [19] have been proposed recently.

In the past, conventional MAC protocols were designed under the assumption that the energy of wireless sensors is unlimited or the energy depletion problem is negligible. However, the energy shortage in a wireless powered sensor brings a significant impact on the system performance of WPSNs [20]. Therefore, novel energy-aware MAC protocols for WPSNs have to be carefully designed by considering the energy statuses of wireless sensors. The most famous protocol for WPSNs is the harvest-then-transmit protocol [21,22,23,24] that each sensor first harvests energy from the access points (APs) and then transmits its data to APs. The harvest-then-transmit protocol uses the time division duplex (TDD) technique which includes the downlink (DL) phase for energy transfer and the uplink (UL) phase for data transmission. The key issue in the harvest-then-transmit protocol is how to properly allocate the time ratio between energy transfer and data transmission in order to maximize the system throughput [25,26]. In the UL phase, the time division multiple access (TDMA) [3,26,27], ALOHA [24], carrier sense multiple access (CSMA) [28], or non-orthogonal multiple access (NOMA) [29] can be employed. For example, the throughput maximization problem was studied in [3,26,27] for the wireless powered communication system which uses TDMA to resolve the contention problem existing in data transmission. The energy efficiency maximization problem in the WPSN for water-quality monitoring was considered in [29], where the NOMA protocol is used for data communication.

In addition to optimizing the time allocation for the UL and DL phases, the MAC protocol for data transmission must be energy-aware as well in order to maximize the system performance. Therefore, the authors in [28] designed an RF-MAC protocol to optimize energy delivery to wireless sensors and minimize disruption to data transmission. RF-MAC is based on the carrier sense multiple access with collision avoidance (CSMA/CA) and includes the following mechanisms: determining the maximum energy charging threshold, choosing specific energy transmitters for charging, requesting and granting power transfer requests, and computing the respective priorities of data transmission and power transfer. In RF-MAC, the sensor with more residual energy gains a higher access priority for data transmission. Another novel harvest-then-transmit approach was designed in [30], where Zheng et al. proposed an energy threshold scheme that jointly considers geographic locations and energy statuses of wireless sensors. In their proposed approach, the wireless power transfer begins when none of wireless sensors has energy over the threshold; otherwise, a randomly selected wireless sensor starts its data transmission. In [31], the authors presented an on-demand MAC protocol (ODMAC) for WPSNs to support individual duty cycles for sensors with different energy profiles. The data transmission occurs in an on-demand manner, i.e., the sensor does not transmit data unless the AP or coordinator asks for it. The paper [32] proposed a MAC protocol for RF-based WPSNs to adaptively manage the sensors’ duty cycles for achieving fairness among them according to their amount of harvested energy and contention time. The work [33] proposed two MAC schemes, including an exponential decision MAC based on current residual energy (ED-CR) and an exponential decision MAC based on prospective increase in residual energy (ED-PIR), by taking the residual energy of each sensor into consideration. Their simulation results show that ED-PIR and ED-CR outperform the static receiver-initiated MAC (RI-MAC) proposed in [34].

However, some advanced technologies such as the beamforming multi-antenna energy transmitters are rarely considered in the aforementioned works [21,22,23,24,25,26,27,28,30,31,32,33]. When beamforming multi-antenna energy transmitters are considered, APs can assign some beamforming antennas to transfer energy to some wireless sensors in a specific direction, yielding an increase in the energy transfer efficiency. Next, most of the energy-aware MAC protocols presented in the literature, e.g., [27,28,30,31,32,33], only consider the residual energy in wireless sensors without taking the data backlog into consideration. To further improve system performance of WPSNs, both the energy status and data backlog of a wireless sensor must be considered simultaneously. Therefore, the beamforming multi-antenna energy transmitter was used in [35], and the residual energy along with the data queue length of each wireless sensor were concurrently considered to optimize the system performance. Note that both TDMA and transmission probability-based random access protocols for resolving the multiple access problem were studied in [35]. However, the work [35] adopted the TDD technique to implement data and energy transfer, i.e., in each time slot the selected wireless sensor first transmits its data packet and then harvests energy from the AP. It means that no two wireless sensors can harvest energy at the same time, implying that the energy transfer efficiency in [35] may not be good.

Notably, one serious flaw of the popular TDD technique or the harvest-then-transmit protocol is that the wireless sensors far apart from the AP are impossible to harvest similar energy as the ones near apart from the AP because of signal attenuation and a comparable charging time for all wireless sensors. Although far-apart wireless sensors need more transmit power for transmitting data in order to achieve a comparable bit error rate at the AP as the near-apart ones do, leading to the so-called doubly near–far problem [7,8]. Therefore, we aim to propose an on-demand energy transfer protocol in this paper for enhancing the energy transfer efficiency and alleviating the doubly near–far problem. In addition, we intend to design an energy-aware polling protocol which determines the polling priority of each wireless sensor based on its residual energy and data backlog to improve the system performance. The contributions of this paper are threefold. First, we propose a frequency division duplex (FDD)-based system architecture for WPSNs to simultaneously support data and energy transfer. Second, an energy-aware polling-based MAC protocol is designed to improve the performance of WPSNs. Finally, an on-demand energy transfer protocol based on the FDD and beamforming techniques is designed to increase the energy transfer efficiency and alleviate the doubly near–far problem encountered in WPSNs.

The rest of this paper is organized as follows. First, the acronyms used in this paper are defined in Table 1. Section 2 describes the system model and assumptions. Section 3 introduces the proposed on-demand energy transfer protocol and energy-aware polling-based MAC protocol. Section 4 conducts simulations to evaluate the performance of our proposed protocols with comparison to the closely related protocols. Finally, the concluding remarks are given in Section 5.

## 2. System Model

The considered system includes a hybrid AP (HAP) with multiple beamforming antennas and *N* wireless powered sensors. The HAP adopts the FDD technique to simultaneously support energy transfer and data transmission. According to the study in [18], FDD significantly improves the spectral efficiency as compared with TDD. Therefore, in this paper we assume that wireless sensors have at least two antennas for supporting the FDD function, i.e., simultaneously harvesting energy and transmitting data. Nowadays, many wireless devices have been implemented with the multiple input multiple output (MIMO) system by using multiple antennas, implying that there is no difficulty to realize two antennas in a wireless sensor. However, the cost of a wireless sensor under FDD may increase as compared with that under TDD. Additionally, energy transfer and data transmission may proceed simultaneously in an FDD system. Thus, energy transfer may interfere with data transmission if their frequency bands are not properly spaced. In this paper, energy is discretized, i.e., divided into energy packets (EPs) of constant size [36]. Each wireless sensor has a data buffer of size *Q* (in data packets) and an energy storage buffer of capacity *C* (in energy packets). The system model and queueing model of a wireless sensor are depicted in Figure 1.

In this paper, an on-demand energy transfer protocol is proposed to improve the energy transfer efficiency and alleviate the doubly near–far problem. When the energy of a wireless sensor is below a given threshold, e.g., less than the required amount of energy for transmitting the head-of-line data packet, it requests the HAP for energy transfer. We assume that the number of beamforming antennas for energy transfer is larger than that of wireless sensors so that no charging request is rejected. In the harvesting energy period, EPs arrive at the *i*-th wireless sensor according to a Poisson process with arrival rate $e_{i}$. When the energy storage buffer of the *i*-th wireless sensor is full, the *i*-th wireless sensor informs the HAP of stopping energy transfer to it. The data packets (DPs) generated by the *i*-th wireless sensor also conform to a Poisson process with rate $\lambda_{i}$. The transmission time of a DP at the *i*-th wireless sensor is assumed to be exponentially distributed with mean $1/\mu_{i}$. At the *i*-th wireless sensor, the energy consumption of transmitting a DP is proportional to the size of a DP and the average number of EPs consumed by transmitting a DP is $r_{i}$. In other words, the energy consumption rate of the *i*-th wireless sensor is $r_{i}\mu_{i}$ in the data transmission phase. In addition, we assume that the energy consumption of wireless sensors is mainly brought by data transmissions.

## 3. Energy-Aware Polling-Based MAC and On-Demand Energy Transfer Protocol

In WPSNs, the MAC protocol for data transmission can be TDMA [26,27], ALOHA [24], CSMA [28], polling [31], etc. ALOHA and CSMA are random access protocols with possible collisions, leading to low bandwidth efficiency in a heavy load condition. TDMA and polling are collision-free protocols. However, TDMA is a static resource allocation protocol and has low bandwidth efficiency in a light load condition. As for the polling protocol, the HAP polls wireless sensors one by one according to their polling priorities, resulting in an additional polling overhead/latency. Due to the fact that the propagation delay in WPSNs is usually negligible and the number of sensors can be constrained to small by constructing multiple HAPs, the impact of polling latency on the system performance will be minor. Another advantage of using multiple HAPs is that multi-source energy transfer schemes can be easily applied to increase the wireless charging rate. Accordingly, in this paper the proposed energy-aware MAC protocol is based on the polling mechanism. This paper aims at designing an efficient and energy-aware polling protocol by determining the polling priorities of wireless sensors based on the numbers of DPs and EPs.

Denote the numbers of DPs and EPs of a wireless sensor to be *x* and *y*, respectively. Then, the polling priority p(x,y) of a wireless sensor is defined as follows:(1)$$p\left(x,y\right)=x{\left(y\right)}^{*},$$
where ${\left(y\right)}^{*}=y$ if *y* is not less than the required amount of energy for transmitting the head-of-line DP; otherwise, ${\left(y\right)}^{*}=0$. Such a polling scheme is called the composite energy and data first (CEDF) protocol. According to (Equation 1), the wireless sensor with more EPs or DPs has a higher polling priority so that it can be polled earlier for data transmission. Notably, when a wireless sensor with insufficient energy is polled, it cannot transmit its DP, leading to an additional polling overhead. To reduce the polling overhead, a wireless sensor with more EPs must be polled earlier. Additionally, a wireless sensor with more DPs must be polled earlier as well in order to shorten its data queue length and delay. Consequently, the product of the number *x* of DPs and the number *y* of EPs is used in Equation (Equation 1) as the polling priority. The HAP polls wireless sensors according to their polling priorities, i.e., p(x,y), in a nonincreasing order. In the proposed polling protocol, the operation procedure at the HAP is illustrated in Figure 2. According to Figure 2, the HAP updates the states of wireless sensors whenever it receives data or p(x,y) reports. Next, the HAP sorts p(x,y) and polls the wireless sensor with the maximum $p^{*}\left(x,y\right)$. If $p^{*}\left(x,y\right)>0$, the polled wireless sensor transmits data; otherwise, the HAP just waits for p(x,y) reports. Based on Figure 2, the computational complexity at the HAP mainly depends on the sorting algorithm which can be performed with the complexity $\Theta (N\log_{2}N)$. As for the computational complexity at the wireless sensor, only one multiplication is needed to derive the polling priority p(x,y). Hence, the complexity at the wireless sensor is very low in the proposed method.

The polling messages include the polling beacon from the HAP to wireless sensors and the polling response from wireless sensors to the HAP. The polling beacon includes at least two fields: the identity (ID) field, indicating the ID of a polled wireless sensor, and the reservation field for each wireless sensor. As for the polling response, it includes at least the reservation field for wireless sensors (two bits for each wireless sensor), as shown in Figure 3. For simplicity, the other fields such as the preamble for synchronization are omitted in Figure 3. When a wireless sensor receives a polling beacon from the HAP, it can request charging by setting its reservation value to 1 in the corresponding bit of the polling response message and sending it to the HAP. When a wireless sensor wants to stop charging, it can set the reservation value to 0 and send it to the HAP. The HAP always notifies wireless sensors of their charging statuses in each polling beacon. When a wireless sensor is polled by the HAP, i.e., receiving a polling beacon with its ID, it then immediately transmits its DP right after the entire polling response message ends.

Explicitly, each wireless sensor must notify the HAP of its polling priority p(x,y) whenever it changes. This can be implemented by estimating the polling priority p(x,y) right after transmitting a DP and reporting the updated polling priority p(x,y) using the piggyback approach. For the wireless sensor not being polled for data transmission, it can first make a reservation for reporting its polling priority p(x,y) by setting the reservation value to 2 or 3, depending on its charging status, in the next polling response message. When the HAP receives a reservation reported by a wireless sensor, the HAP will poll this wireless sensor as soon as possible. Therefore, the polling priority p(x,y) reporting has a higher transmission priority than ordinary DPs. The operation of our proposed polling protocol is illustrated by Figure 4.

## 4. Simulation Results and Discussion

In this section, we conduct numerical examples to evaluate the performance of the proposed energy-aware polling protocol CEDF under the proposed on-demand energy transfer protocol. The wireless sensors are uniformly distributed over a square area of 20 × 20 $m^{2}$ while the HAP is located at the center of the square, as shown in Figure 5. The number of wireless sensors is denoted by *N*. The data buffer size and energy storage capacity of each wireless sensor are *Q* DPs and *C* EPs, respectively. The polling latency is denoted by τ ms, as shown in Figure 4. The arrival processes of DPs and EPs in the *i*-th wireless sensor are Poisson processes with arrival rates $\lambda_{i}$ (1/ms) and $e_{i}$ (1/ms), respectively. The transmission time of a DP at the *i*-th wireless sensor is exponentially distributed with service rate $\mu_{i}$ (1/ms). At the *i*-th wireless sensor, the energy consumption of transmitting a DP is proportional to the transmission time of a DP and its average is $r_{i}$ EPs. Additionally, at each wireless sensor the energy leakage and energy consumption of transmitting the polling response are ignored [35]. Notably, the average wireless charging rate $e_{i}$ conforms to the Friis transmission formula or power law, i.e.,
(2)$$e_{i}\propto \frac{P_{t}}{d_{i}^{2}},$$
where $P_{t}$ is the transmit power of a beamforming antenna at the HAP and $d_{i}$ is the distance between the HAP and the *i*-th wireless sensor. In this section, we assume that, at most, one beamforming antenna is allocated to each wireless sensor for energy transfer. The system traffic load ρ is defined by $\rho =\sum_{i=1}^{N}\lambda_{i}/\mu_{i}$. The simulation programs are developed by ourself using C++ Builder. For each data point in the following figures, five simulation runs are conducted and the average of all measures is presented. Each performance measure comes from the average of all the wireless sensors. To demonstrate the advantages of the proposed CEDF polling protocol, the performance of the conventional round robin (RR) and data first (DF) polling protocols is accompanied for comparison. In the RR polling protocol, the HAP polls all the wireless sensors sequentially according to a preset order, e.g., in the increasing order of IDs. In the DF polling protocol, the wireless sensor with the most DPs is polled first, i.e., its polling priority is defined as p(x)=x.

In the first numerical example, the parameters are set as follows: N=10, C=300, Q=10, τ=0.4, and $\mu_{i}=1.25$ for all *i*. The DP arrival rate of each wireless sensor is the same, i.e., $\lambda_{i}=\lambda$. As for the wireless charging rate $e_{i}$, it is set to $1000/d_{i}^{2}$ according to (Equation 2). To preserve a comparable bit error rate at the HAP for all wireless sensors, the transmit power of the *i*-th wireless sensor must be proportional to $d_{i}^{2}$. Thus, the energy consumption of the *i*-th wireless sensor during transmitting a DP is also proportional to $d_{i}^{2}$. Consequently, the average number of EPs required for transmitting a DP at the *i*-th wireless sensor is assumed to be $r_{i}=\left\lceil d_{i}^{2}\right\rceil$. Figure 6 shows the DP loss rate versus the system traffic load ρ by varying the DP arrival rate λ under the RR, DF, and CEDF polling protocols. Obviously, the proposed CEDF significantly outperforms RR and DF, especially in heavy load conditions. Since the proposed CEDF has the lowest DP loss rate, it achieves the highest throughput as compared with RR and DF, as shown in Figure 7. The corresponding average DP delay is plotted in Figure 8, where CEDF achieves the lowest delay among the three polling protocols. From Figure 6, Figure 7 and Figure 8, one can tell that DF performs worst while the proposed CEDF performs best. The reasons are explained as follows. Because CEDF further takes the energy statuses of wireless sensors into consideration, the HAP never polls a wireless sensor with insufficient energy for transmission, yielding the least polling overhead and the best system performance. Under the DF protocol, the wireless sensor with more DPs is more frequently polled so that more energy is consumed, leading to a higher energy shortage probability accordingly. It means that the HAP more frequently polls the wireless sensor with a higher energy shortage probability, resulting in a higher polling overhead as compared with RR and CEDF. Thus, DF achieves the worst system performance.

Figure 9 shows the charging time ratio versus the system traffic load ρ under RR, DF, and CEDF. Under the proposed on-demand energy transfer protocol, the charging time ratio is defined to be the ratio of total charging time to overall simulation time. From Figure 9, CEDF has the largest charging time ratio and, thus, harvests the most energy as compared with RR and DF. Undoubtedly, in the CEDF protocol the wireless sensor can transmit more DPs because more energy is harvested as compared with RR and DF, and, thus, CEDF achieves the highest throughput. Additionally, the energy transfer can continuously proceed as wireless sensors need when the FDD-based on-demand energy transfer protocol is used. However, in the conventional TDD-based harvest-then-transmit protocol the charging time of wireless sensors is similar, i.e., without considering diverse charging requirements of wireless sensors and lack of flexibility. In Figure 9, the charging time ratio is always less than 50% for all the polling protocols, demonstrating that the energy transfer is flexible and highly efficient under the proposed on-demand energy transfer protocol.

Next, let us investigate the doubly near–far problem in WPSNs under the proposed CEDF along with the on-demand energy transfer protocol. The doubly near–far problem says that the wireless sensors far apart from the HAP hardly harvest the energy while they need more energy for transmitting DPs in order to preserve the same bit error bit at the HAP as those near apart from the HAP do. The performance of the three wireless sensors among 10 wireless sensors in the previous example is studied. The related information and parameters of the three selected wireless sensors (denoted by A, B, and C specifically) are summarized in Table 2.

Figure 10 and Figure 11 indicate the DP loss rate and average delay performance of wireless sensors A, B, and C. Obviously, the wireless sensor farther away from the HAP has much worse performance than the one nearer away from the HAP. Figure 12 shows the charging time ratios at wireless sensors A, B, and C. From this Figure, one can see that wireless sensor C which is farthest away from the HAP has the largest charging time ratio. In Figure 12, the charging time at a wireless sensor increases as the distance between it and the HAP, demonstrating that the proposed on-demand energy transfer protocol is very flexible and able to meet different charging requirements of wireless sensors. Although wireless sensor C has the longest charging time under the on-demand energy transfer protocol, it still exhibits the worst performance among wireless sensors A, B, and C. This is because the charging rate and energy depletion rate of a wireless sensor are proportional to $1/d_{i}^{2}$ and $d_{i}^{2}$, respectively. This phenomenon is the so-called doubly near–far problem encountered in WPSNs.

Finally, we intend to find solutions for solving the doubly near–far problem in WPSNs through examining the performance of wireless sensors A, B, and C versus the transmit power $P_{t}$ at the HAP. The relationship between the charging rate $e_{i}$ and the transmit power $P_{t}$ is given by Equation (Equation 2). Figure 13 and Figure 14 indicate that the DP loss rate and average delay, respectively, by varying the transmit power $P_{t}$ at the HAP. Figure 15 presents the charging time ratios of wireless sensors at different locations versus the transmit power $P_{t}$ at the HAP. By Figure 13 and Figure 14, the performance of wireless sensors is significantly improved when the transmit power $P_{t}$ is increased. Additionally, at a low transmit power $P_{t}$, the doubly near–far phenomenon is very obvious, while it gradually alleviates as the transmit power $P_{t}$ grows and, finally, all the wireless sensors perform comparably at a large $P_{t}$. The required charging time at wireless sensors also decreases significantly at a large transmit power $P_{t}$. All these results demonstrate that the doubly near–far problem can be effectively alleviated by increasing the transmit power $P_{t}$ at the HAP for energy transfer. However, the transmit power $P_{t}$ of a power beam is impossible to be increased unlimited because of the human health reason. Hence, other effective energy transfer techniques, such as the multi-source energy transfer scheme for improving the charging rate while assuring the health condition, must be further studied in the future.

## 5. Conclusions

In this paper, we propose an energy-aware polling-based MAC protocol CEDF for WPSNs adopting the on-demand energy transfer protocol. The proposed FDD-based on-demand energy transfer protocol, where a wireless sensor requests for charging when its energy is insufficient and requests for stopping charging when its energy storage buffer is full, can effectively improve the energy transfer efficiency and alleviate the doubly near–far problem. The proposed CEDF polling protocol determines the polling priority of a wireless sensor according to its numbers of DPs and EPs. Simulation results show that the proposed CEDF protocol outperforms the other conventional polling protocols, such as RR and DF, in terms of the DP loss rate, throughput, and DP delay. Additionally, the doubly near–far problem in WPSNs has been investigated as well. Simulation results indicate that increasing the transmit power at the HAP or using multi-source energy transfer scheme to increase the wireless charging rate can effectively alleviate the doubly near–far problem. In the near future, we will construct a multi-source multi-sensor WPSN testbed as in [10] for testing the proposed on-demand energy transfer and polling protocol. 

## Figures and Tables

**Figure 1 sensors-22-02476-f001:**
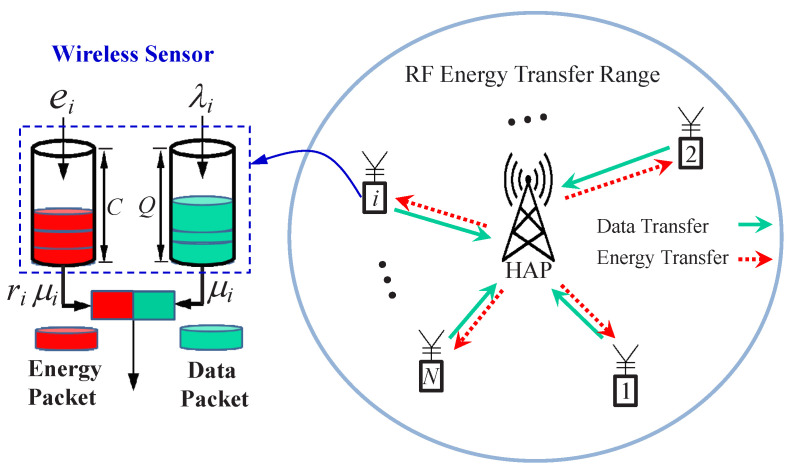
System model and queueing model of a wireless sensor.

**Figure 2 sensors-22-02476-f002:**
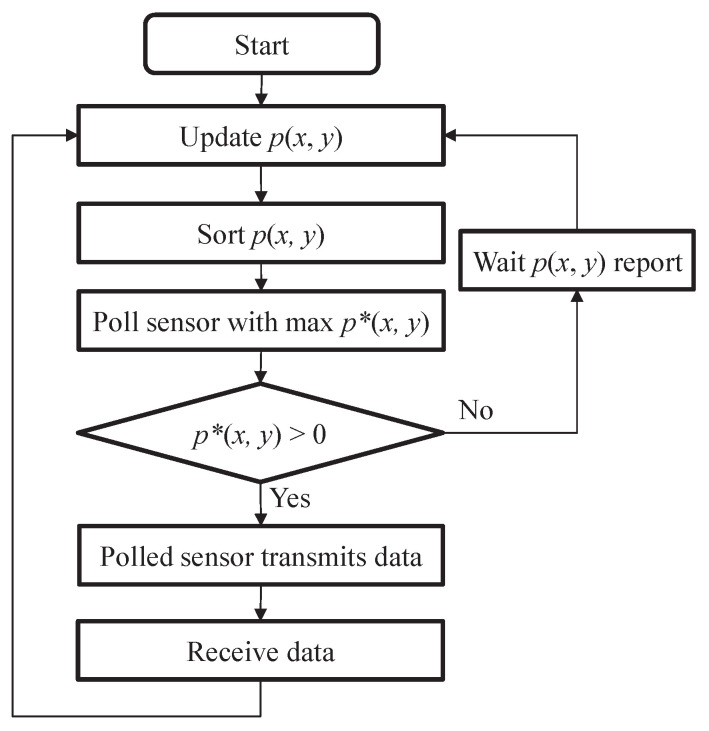
Operation procedure at the HAP under the proposed polling protocol.

**Figure 3 sensors-22-02476-f003:**
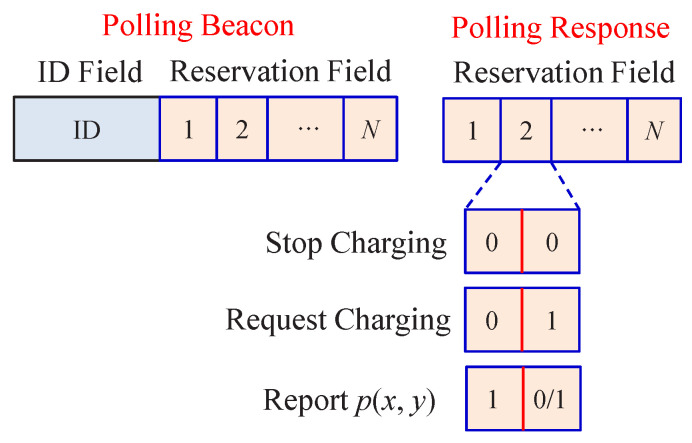
Message formats of polling beacon and response.

**Figure 4 sensors-22-02476-f004:**
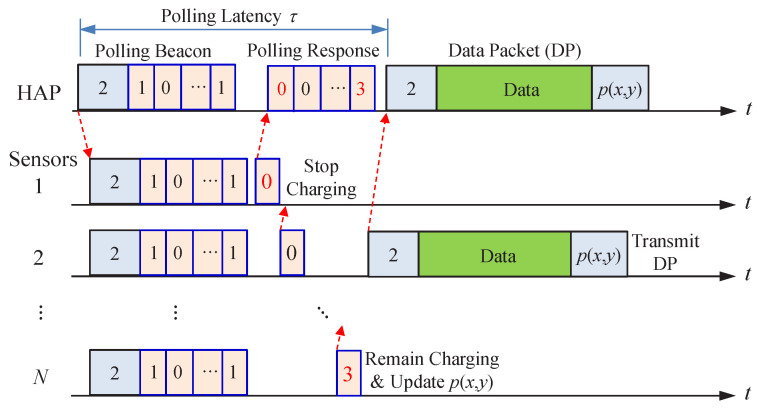
Illustration of the energy-aware polling protocol.

**Figure 5 sensors-22-02476-f005:**
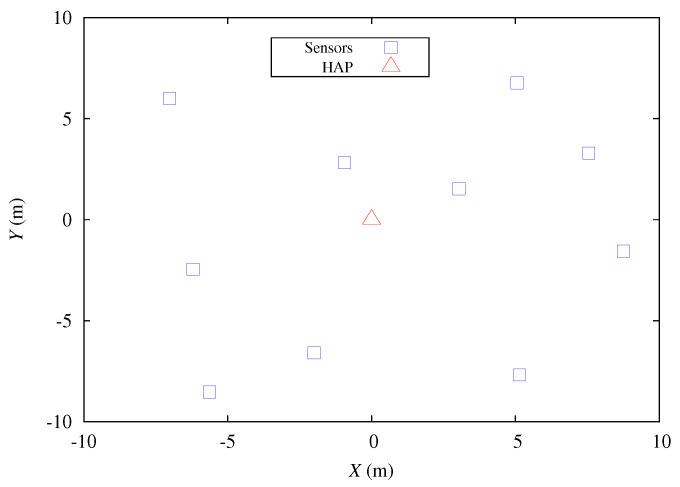
Deployment of HAP and wireless sensors in simulations. (N=10).

**Figure 6 sensors-22-02476-f006:**
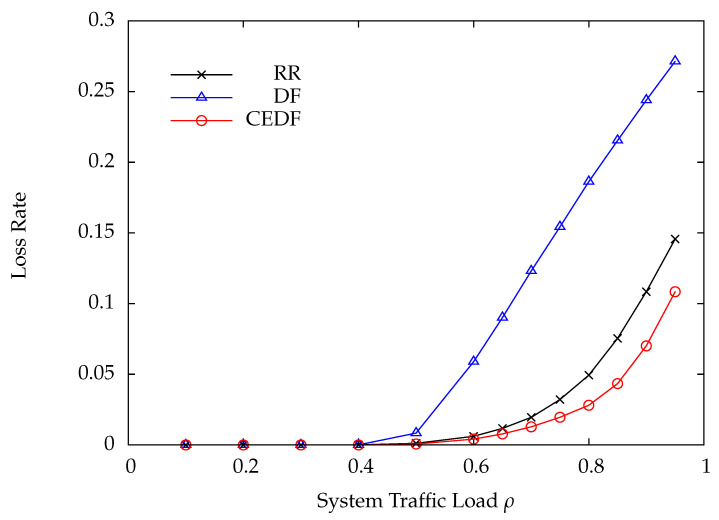
Loss rate versus system traffic load ρ.

**Figure 7 sensors-22-02476-f007:**
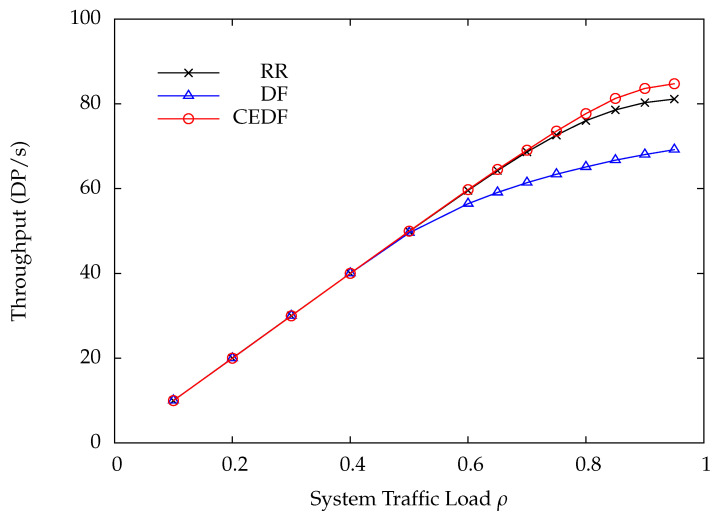
Throughput versus system traffic load ρ.

**Figure 8 sensors-22-02476-f008:**
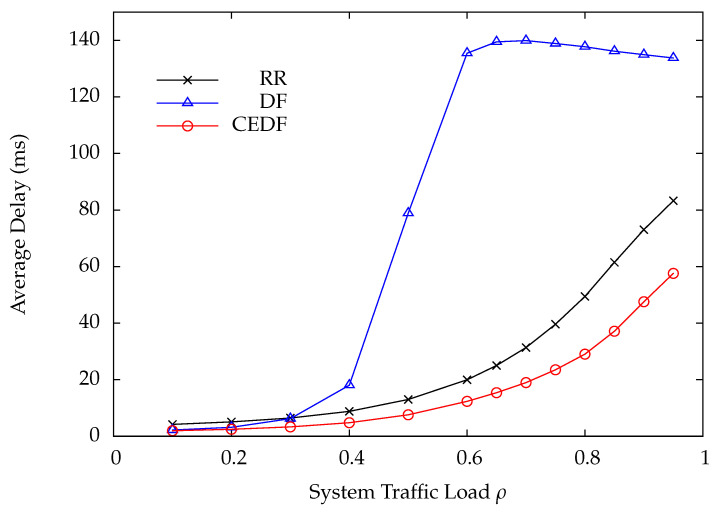
Average delay versus system traffic load ρ.

**Figure 9 sensors-22-02476-f009:**
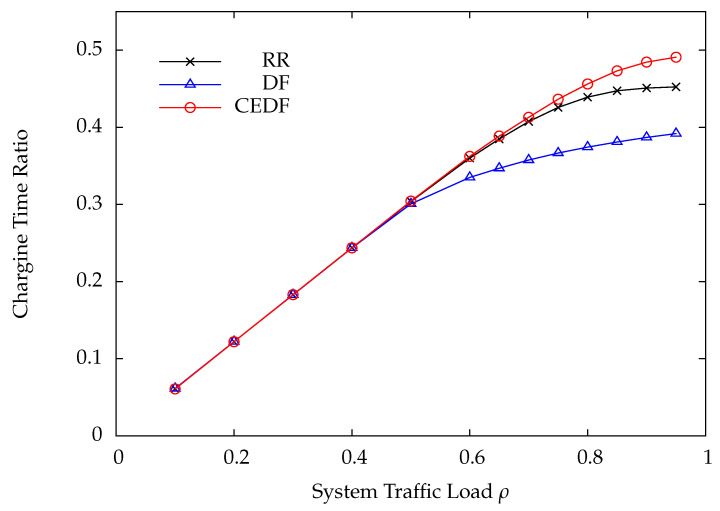
Charging time ratio versus system traffic load ρ.

**Figure 10 sensors-22-02476-f010:**
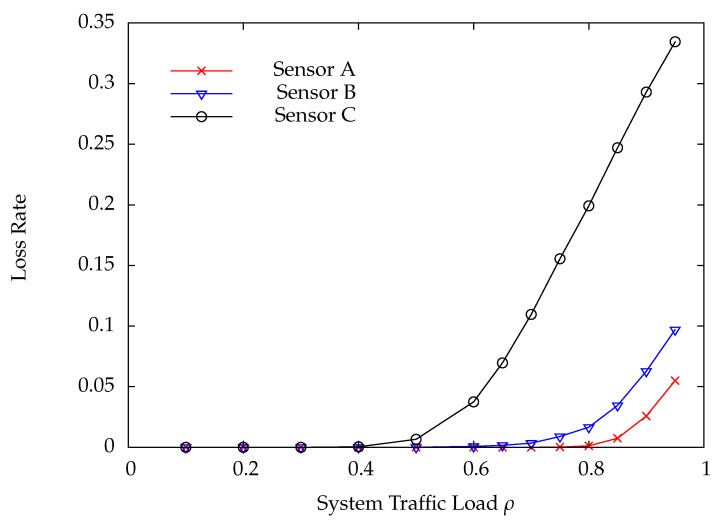
Loss rate versus system traffic load ρ.

**Figure 11 sensors-22-02476-f011:**
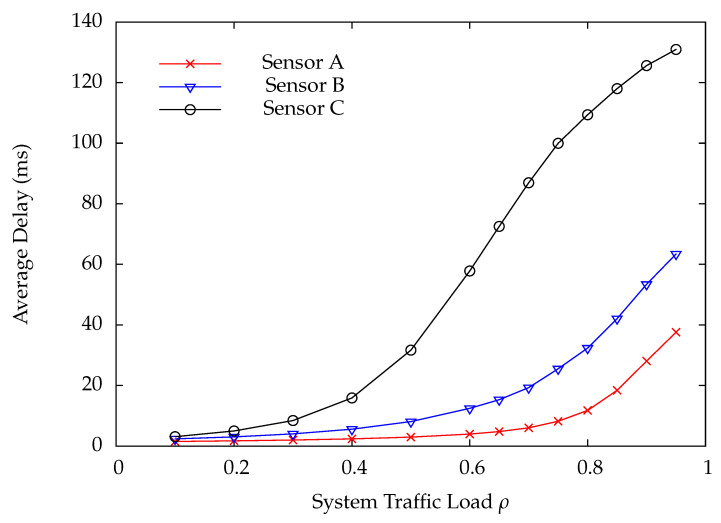
Average delay versus system traffic load ρ.

**Figure 12 sensors-22-02476-f012:**
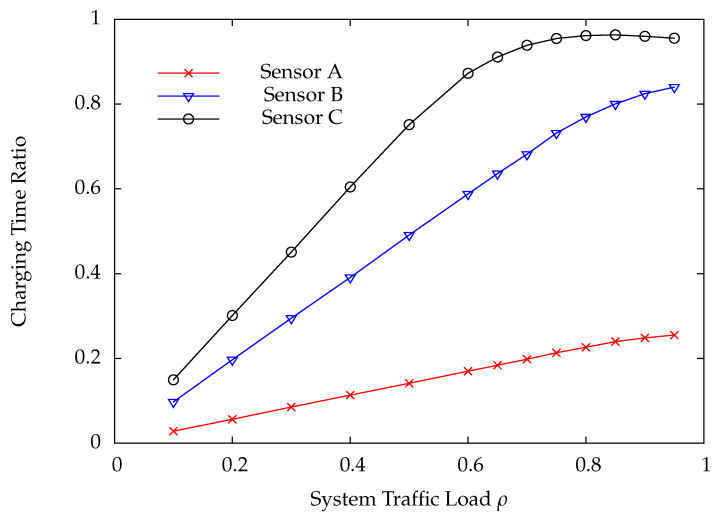
Charging time ratio versus system traffic load ρ.

**Figure 13 sensors-22-02476-f013:**
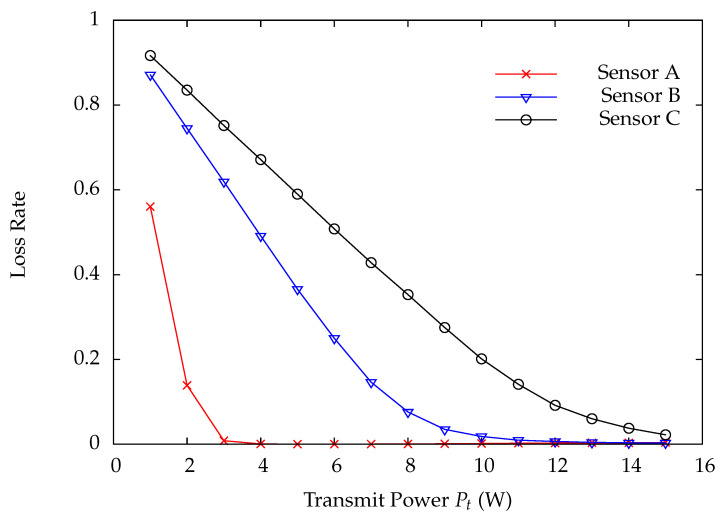
Loss rate versus transmit power $P_{t}$.

**Figure 14 sensors-22-02476-f014:**
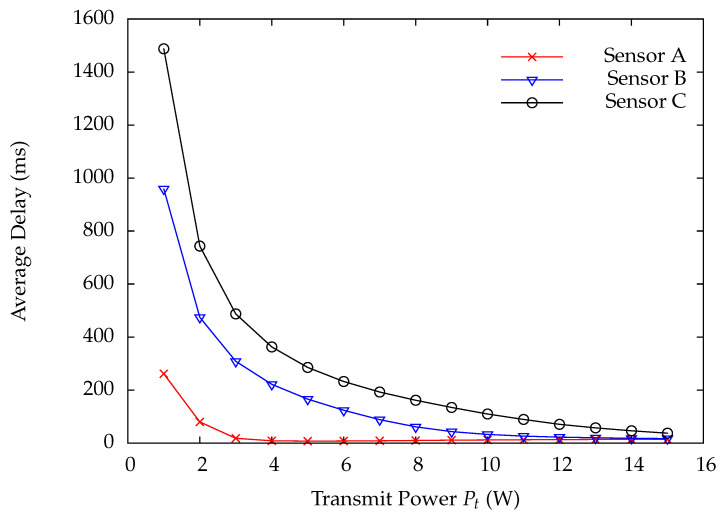
Average delay versus transmit power $P_{t}$.

**Figure 15 sensors-22-02476-f015:**
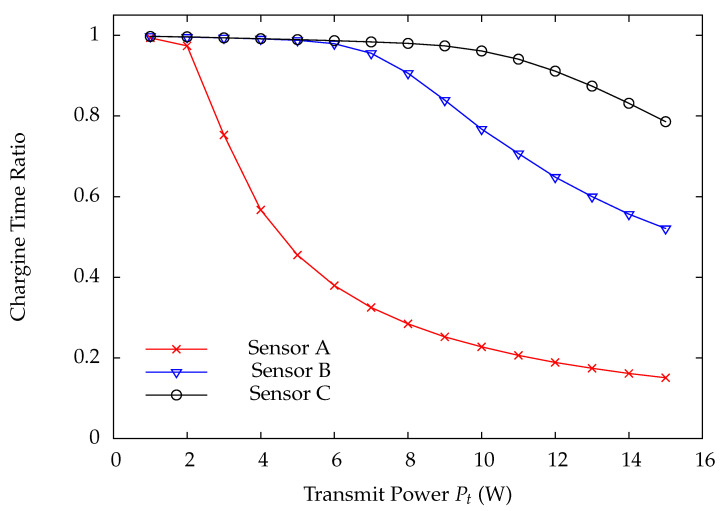
Charging time ratio versus transmit power $P_{t}$.

**Table 1 sensors-22-02476-t001:** Listing of acronyms with description.

Acronym	Description	Acronym	Description
RF	Radio Frequency	WPSN	Wireless Powered Sensor Network
IoT	Internet of Things	MAC	Medium Access Control
AP	Access Point	MIMO	Multiple Input Multiple Output
HAP	Hybrid Access Point	FDD	Frequency Division Duplex
DL	Downlink	TDD	Time Division Duplex
UL	Uplink	TDMA	Time Division Multiple Access
DF	Data First	CEDF	Composite Energy and Data First
RR	Round Robin	NOMA	Non-Orthogonal Multiple Access
DP	Data Packet	CSMA	Carrier Sense Multiple Access
EP	Energy Packet	CSMA/CA	Carrier Sense Multiple Access/Collision Avoidance

**Table 2 sensors-22-02476-t002:** Parameters of wireless sensors A, B, and C.

Sensor	Distance $d_{i}$ (m)	Charging Rate $e_{i}$ (EP/ms)	Energy Consumption $r_{i}$ (EP)
A	7.29	18.816	52
B	10.06	9.881	102
C	11.35	7.762	129

## Data Availability

Not applicable.

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
