# Peer review of "On-Demand Energy Transfer and Energy-Aware Polling-Based MAC for Wireless Powered Sensor Networks"

_sensors, 2022, doi:10.3390/s22072476_

Round 1

Reviewer 1 Report

The problem being addressed in this paper is interesting.  Below listed some questions and comments. 

  • (Line 100) The advantage of FDD is described. Any disadvantage compared to TDD?
  • (Line 115) Any reason for assuming that the transmission time of DP is exponentially distributed?
  • (Line 82) The authors mentioned "doubly near-far problem". Any special care is not needed in terms of polling priority for such far-apart sensors?

Reviewer 2 Report

Comment:

I would like to commend the authors for putting this manuscript together. The authors have proposed the development of a polling-based MAC protocol for addressing energy transfer and data transmission in wireless powered sensor networks. The manuscript is well written. The text is clear and easy to read. The conclusions of the manuscript are consistent with the evidence and arguments presented. Also, the authors have made some efforts to address the main question posed. The manuscript has a potential for publication after a major revision. However, the following issues must be addressed to improve the quality of this manuscript:

(1) The authors need to provide an algorithm that describe how the proposed method compute resources for the wireless sensors in the proposed system.
(2) Due to the inherent limited power of the water sensors, the authors are required to analyze the complexity of the proposed method.
(3) The authors should discuss plans for the real-life testing of the proposed solution.
(4) Most of the figures in the manuscript are unclear, hence, authors need to improve on their quality.
(5) The manuscript lacks important and sufficient references on wireless powered sensor networks and radio frequency power transfer. Hence, the authors are advised to consider the following works in the introduction section.

-Radio frequency energy harvesting and data rate optimization in wireless information and power transfer sensor networks, vol. 17, no. 15, 2017.
-Optimizing the energy and throughput of a water-quality monitoring system. Sensors (MDPI), vol. 18, no. 4, 2018.
-Efficient energy resource utilization in a wireless sensor system for monitoring water quality. EURASIP Journal on Wireless Communications and Networking, vol. 6, 2019.
-Sum-throughput maximization in wireless sensor networks with radio frequency energy harvesting and backscatter communication. IEEE Sensors Journal, vol. 18, no. 17, 2018.
-Energy efficiency maximization in a wireless powered IoT sensor network for water quality monitoring. Computer Networks (Elsevier), vol. 176, 2020.

Reviewer 3 Report

The paper presents a long standing isuue on power wirelesss sensor networks, and the idea of combining existed well known techniques and protocols into creating a newly proposed CEDF is orchestrated nicely and with good suceess.

My only objection is that the simulation experiments are performed in rather ideal network conditions. I do not think the scalability issue is explored and how the system performs in large scale networks.

I suggest to carefully pass the paper and correct o few minor English language corrections.

I would also suggest to include a list of acronyms.

I would suggest a careful pass of the paper and correct some minor english problem.

Round 2

Reviewer 1 Report

All the issues raised in the previous round of review have been addressed in the revised manuscript. 

Reviewer 2 Report

Comments:

I confirm that the revised manuscript has been satisfactorily updated and is now suitable for publication.